# Proposal of New Health Risk Assessment Method for Deficient Essential Elements in Drinking Water—Case Study of the Slovak Republic

**DOI:** 10.3390/ijerph17165915

**Published:** 2020-08-14

**Authors:** Stanislav Rapant, Veronika Cvečková, Edgar Hiller, Dana Jurkovičová, František Kožíšek, Beáta Stehlíková

**Affiliations:** 1Department of Geochemistry, Faculty of Natural Sciences, Comenius University in Bratislava, Ilkovičova 6, 842 15 Bratislava, Slovakia; veronika.cveckova@uniba.sk (V.C.); edgar.hiller@uniba.sk (E.H.); 2Cancer Research Institute, Biomedical Research Center, Slovak Academy of Sciences, Dúbravská cesta 9, 845 05 Bratislava, Slovakia; danus.jurkovic@gmail.com; 3National Institute of Public Health, Šrobárová 49/48, 100 00 Praha 10, Czech Republic; frantisek.kozisek@szu.cz; 4Faculty of Economics of Business, Pan-European University, Tematínska 10, 851 05 Bratislava 5, Slovakia; stehlikovab2@gmail.com

**Keywords:** human health risk assessment, deficiency, essential elements, drinking water, calcium, magnesium, average daily missing dose

## Abstract

The US EPA health risk assessment method is currently widely used to assess human health risks for many environmental constituents. It is used for risk assessment from the exposure to various contaminants exceeding tolerable or safe reference doses, determined e.g., for drinking water, soil, air and food. It accepts widely that excess contents of non-essential elements (e.g., As, Pb or Sb) in environmental compartments represent a general risk to human health. However, contrary to toxic trace elements, deficient contents of essential (biogenic) elements e.g., F, I, Se, Zn, Fe, Ca or Mg may represent even higher health risk. Therefore, we propose to extend the human health risk assessment by calculating the health risk for deficient content and intake of essential elements, and to introduce the terms Average Daily Missing Dose (ADMD), Average Daily Required Dose (ADRD) and Average Daily Accepted Dose (ADAD). We propose the following equation to calculate the Hazard Quotient (HQ) of health risk from deficient elements: HQ_d_ = ADRD/ADAD. At present, there are no reference concentrations or doses of essential elements in each environmental compartment in world databases (Integrated Risk Information System IRIS, The Risk Assessment Information System RAIS). ADRD and ADMD can be derived from different regulatory standards or guidelines (if they exist) or calculated from actual regional data on the state of population health and content of essential elements in the environment, e.g., in groundwater or soil. This methodology was elaborated and tested on inhabitants of the Slovak Republic supplied with soft drinking water with an average Mg content of 5.66 mg·L^−1^. The calculated ADMD of Mg for these inhabitants is 0.314 mg·kg^−1^·day^−1^ and HQ_d_ is equal to 2.94, indicating medium risk of chronic diseases. This method extending traditional health risk assessment is the first attempt to quantify deficient content of essential elements in drinking water. It still has some limitations but also has potential to be further developed and refined through its testing in other countries.

## 1. Introduction

The health risk represents the probability of damage, illness or human death as a result of the environmental risk factor effect [1]. Health risk assessment from a contaminated natural environment is therefore an important tool from the point of view of protecting the health of human populations, and in a broader context, also from the point of view of maintaining sustainable development. The health risk assessment methodology was formulated in the 1980s by the US Environmental Protection Agency (US EPA)—so called the method of Human Health Risk Assessment [1]. Its main principles apply with some modifications to date [2,3,4,5,6] and have become the basis for the legislative elaboration of health risk assessment procedures within the European Union [7,8] and also in individual countries, e.g., in the Slovak and Czech Republic [9,10].

The formulation of basic concepts and procedures for health risk assessment and their unification at the legislative level currently allows, in addition to a qualitative approach to the assessment of adverse effects of elements/substances on humans, the quantitative determination of risk levels in relation to these effects.

Quantitative determination of health risk is expressed by the hazard quotient (HQ) for non-carcinogenic risk (threshold effect) or as individual lifetime cancer risk (ILCR), and optionally, annual population cancer risk (APCR), expressed in terms of the probability, i.e., number of cases of cancer per capita per year (non-threshold effect).

The current methodological procedures for the health risk calculation only deal with increased contents of harmful substances/elements. They assess possible adverse effects of various substances/elements that occur at contents above the limit, or reference dose. Reference doses of different harmful elements are set for different environmental compartments, and thus, e.g., human health risk from increased As content in soils, air, drinking/groundwater or in various foods (e.g., vegetables) is estimated. However, these procedures do not assess the health risk due to the deficient content of various—especially biogenic—essential elements necessary for healthy human development. A classic example of such elements is deficient content of iodine, fluorine, iron and several trace elements (Se, Zn and Cu) or essential macro-elements e.g., Ca and Mg [11,12,13,14]. Deficient elements are currently added to food (e.g., iodine in table salt, fluorine in toothpaste) or are available in the form of various nutritional supplements (Zn, Se, Fe or others). Important essential elements include Ca and Mg, whose low contents in drinking water are often associated with an increased incidence/mortality from several serious chronic diseases, especially cardiovascular diseases [15,16,17]. There are far fewer studies on cancer, respiratory or gastrointestinal diseases and most of them are only ecological types of studies, so the strength of the evidence is still low. However, there is the growing evidence on the important role of sufficient intake of Ca and Mg in the immune system [18,19,20,21], which may provide explanation on the missing pathological mechanism involved. These relationships are so far known exclusively as the threshold type; therefore, only the concept of non-carcinogenic risk and the calculation of HQ will be worked on.

The aim of the present article is to propose a new methodology for the health risk calculation from the deficit content of the essential elements and to test it on the example of the content of Ca and Mg in drinking water of the Slovak Republic. Previous works using artificial neural network calculations confirmed that Ca and Mg were the elements most affecting the health of the population in the Slovak Republic [22,23]. Their effect is up to two orders of magnitude higher than the effect of classical contaminants in drinking water, such as potentially toxic elements or nitrates. Under their deficit contents, the relative mortality for the main causes of death in the Slovak Republic (i.e., cardiovascular and oncological diseases, diseases of the digestive and respiratory system) increases significantly by 55%–120%. The average life expectancy decreases by up to five years with a deficient content of Ca and Mg. The basis for calculating this type of health risk is to determine the average daily required dose (ADRD), the dose of the element necessary for the healthy development of a person. The ADRD can be derived in two ways. The first way is very simple, using various recommendations or limit values, if any. The second way is much more complicated; it requires a comparison of the health status of populations with various contents of elements in the environment (e.g., drinking water and soil), thus including the use of epidemiological studies. From ADRD, ADMD can be calculated.

The reference dose and similar concepts (tolerable daily intake and acceptable daily intake) work with the total exposure of elements or substances; thus, these are the combined exposure of ingestion, inhalation and dermal exposure, as far as they are relevant for the substance. In our model, we differ because we only consider the intake of elements by drinking water—there is a certain parallel in the assessment of health risks of airborne contaminants, where HQ is calculated as the ratio of the concentration of a substance in the air and the reference concentration for this substance (both quantities in units mg·m^−3^). This important premise, that the effect of insufficient intake of essential elements by drinking water is to some extent independent of food intake of the given element, is derived mainly from animal experiments. They showed that even though animals receive a complete diet, which ensures their need for essential elements for one hundred percent, the low content of these elements in served drinking water will also have a negative effect on their health [24,25]. However, the premise is also indirectly confirmed by epidemiological studies carried out in developed countries. Although there is no problem with malnutrition of the population in developed countries, there are still significant differences in the mortality of the population among localities with different Ca and Mg contents in drinking water.

## 2. Materials and Methods

### 2.1. Area Description

An epidemiological study, needed to derive dose-response characteristics for deficit essential elements in drinking water, was processed in the territory of the Slovak Republic (number of inhabitants: 5.5 million; total area of approximately 50,000 km^2^ (Figure 1)). The Slovak Republic is characterized by a very complicated geological structure [26]. The geological processes of three Wilson’s cycles: Hercinic, Cadon-Caledonian and Alpine, contributed to its development. As a result of these multiple tectonic processes, different rocks of the Paleozoic, Crystalline, Mesozoic, Paleogene and Neogene alternate in a relatively small area. Due to the diversity of the rock environment, the chemical composition of groundwater/drinking water is also highly variable. Residents in individual municipalities are therefore supplied with groundwater/drinking water of different genesis and different chemical composition. There are waters with different mineralization from <50 mg·L^−1^ (silicate groundwater) up to more than 1000 mg·L^−1^, sulphate or sulphate-carbonate waters [27]. For the purpose of this work, two groups of residents were compiled: group of residents supplied with soft drinking water and group of residents supplied with hard drinking water. Data on the chemical composition of drinking water and the health status of the population were available from all municipalities.

### 2.2. Selection of Municipalities

Two groups of municipalities supplied with drinking water of different hardness and different content of Ca and Mg were compiled within the project LIFE-WATER and HEALTH [28]. Municipalities were selected randomly from the entire territory of the Slovak Republic (Figure 1). Each group included more than 50,000 residents. The first group of municipalities, hereafter referred as the “soft” water group, consisted of 34 municipalities supplied with drinking water of low Ca content (<30 mg·L^−1^), Mg content of <10 mg·L^−1^ and low total water hardness (<1.0 mmol·L^−1^) (Table 1). The total population in this group was 52,676. The second group of municipalities, hereafter referred as the “hard” water group, consisted of 21 municipalities with high Ca content (>50 mg·L^−1^), Mg content of >25 mg·L^−1^ and total water hardness (>2.5 mmol·L^−1^) in drinking water (Table 1). The total population in this group was 53,118. In the “soft” water group, a larger number of municipalities was selected because groundwater of silicate environments has a lower yield. In the Slovak Republic, municipalities with several thousand residents supplied with soft drinking water are rare.

There was no change in the source of drinking water in any municipality during the last fifteen evaluated years. Municipalities with a minimum of 500 residents were selected. The numbers of residents in municipalities were not random. It is clear from our previous studies [29,30] that small municipalities below 100 residents often showed unbalanced health indicators (small number error). On the other hand, large municipalities (towns) with ≥10,000 residents are often supplied with drinking water from several sources. We also wanted to avoid the fact that when 4–5 municipalities with >10,000 residents were selected, the necessary variability of health indicators could not be obtained. Municipalities with 500–5000 residents showed the most balanced health indicators and were most closely connected with the environment [29,30]. The selected municipalities are of rural character with similar socio-economic characteristics and other major determinants of health, such as lifestyle, quality and availability of health care [16].

The number of inhabitants, both in the Slovak Republic and selected groups of municipalities, is practically unchanged. The number of inhabitants in the Slovak Republic increased slightly from 5,356,207 in 1995 to 5,443,120 in 2018. A similar, very slight increase was recorded in the “hard” water group (from 53,096 inhabitants in 1995 to 53,992 inhabitants in 2018). On the other hand, there was a decrease in the number of inhabitants (52,718 inhabitants in 1995 and 52,411 inhabitants in 2018) in the “soft” water group. This decrease is caused by the natural migration of the population to larger municipalities or cities [31].

### 2.3. Characteristics of Ca, Mg Content and Water Hardness in Municipalities Supplied with Drinking Water of Different Hardness

According to the Slovak standard for drinking water, the hardness of water is given as the content of Ca + Mg in mmol·L^−1^ [32] and it is expressed in the same way here. The contents of Ca, Mg and water hardness for both studied groups are given in Table 1. It is obvious from Table 1 that there is a fundamental difference in Ca, Mg content and water hardness level between the two water groups. The values of the parameters are 3–4 times lower in the “soft” water group than in the “hard” water group. A detailed overview of the selected municipalities, the basic chemical composition of individual water sources, the number of residents in individual municipalities, etc., is available on the website http://fns.uniba.sk/lifewaterhealth/ [28].

### 2.4. Characteristics of the Health Status of the Population in Municipalities Supplied with Drinking Water of Different Hardness

The health status of residents is given on the basis of health indicators (HIs). Health indicator data for the period of 1994–2008 from the Statistical Office of the Slovak Republic [31] were used. All data thus represent the 15-year average. All HIs were compiled according to the International Classification of Diseases (ICD), 10th revision [33], in accordance with the WHO recommendation [34,35,36]. The method of HI construction and methodology of their calculation is described by Rapant et al. in 2014 [22] or is available in full detail on the web page of project GEOHEALTH [30].

The most common four HIs were selected for the evaluation, which together represent approximately 88% of causes of death in the Slovak Republic: relative mortality from cardiovascular diseases (ReI), relative mortality from oncological diseases (ReC), relative mortality from respiratory diseases (ReJ) and relative mortality from gastrointestinal diseases (ReK). All used HIs are robust, stable, easy to construct and internationally comparable. Data on mortality from individual diseases in comparison with average national values are shown in Table 2. Life expectancy at birth (LE) is also given for illustration. Table 2 shows fundamental differences in HIs between the two selected groups. Health indicators of the “soft” water group are significantly worse (by 50%–100%) than in the “hard” water group. A detailed description of the health status of residents in the two selected groups of municipalities is available at www.uniba.sk/lifewaterhealth/ [28].

### 2.5. Methodology of Hazard Quotient Calculation for Deficient Essential Elements

The basis for calculating the health risk of various contaminants with threshold type of toxicological effect in the sense of the US EPA is the reference dose (RfD), established for individual elements or contaminants in the literature, and the average daily dose (ADD), calculated from the actual element content according to the exposure scenarios designed, i.e., site and population specific. The ratio of ADD to RfD is equal to HQ and its size indicates possible health risk. If the value of HQ is less than 1, negative health effect is not expected; if it is higher than 1 (i.e., ADD is higher than RfD), health damage cannot be excluded, but the higher HQ is, the higher the probability of health damage. However, in our model, the health risk is not caused by excess of harmful elements but deficiency of essential elements. From the human health point of view, Ca and Mg have not been not yet considered by regulatory agencies as risk elements, and therefore, the minimum necessary or maximum permissible daily doses for individual components of the environment are not defined in world toxicological databases. Based on the RfD and ADD, then HQ is calculated according to the US EPA methodology. In our case, the risk does not relate to the excess of harmful elements but the absence of necessary essential elements (of course, any excess of intake, even of essential elements, may be harmful, but it is very rare case regarding Ca or Mg dietary intake). Therefore, to assess human health risk for deficient essential elements, the following kinds of average daily doses (in mg·kg^−1^ bw·day^−1^) are proposed:Average Daily Required Dose (ADRD),Average Daily Accepted Dose (ADAD),Average Daily Missing Dose (ADMD).

The proposed doses are needed to calculate the Hazard Quotient for a deficient essential element (HQ_d_) according to the following equation:HQ_d_ = ADRD/ADAD,(1)
where ADRD represents the average daily dose of an essential element from a given exposure source required for the healthy development of a person; if one does not receive it, he/she is at health risk. Although this quantity is similar to the various nutritional recommendations of the RDA (recommended dietary allowances) type, it is reduced only to exposure from one environmental compartment (here drinking water), as long as the data allow the calculation of such a dose. ADAD represents the average daily intake of an essential element from individual components of the environment, in our case, from drinking water. ADMD represents the average daily missing dose of an essential element.

As already mentioned, the risk analysis methodology for deficient elements was developed and tested for Ca and Mg content in groundwater/drinking water. In the Slovak Republic, groundwater represents approximately 90% of drinking water sources [37], and therefore, groundwater represents drinking water in this study. All selected municipalities are supplied from groundwater sources. When assessing the health risk for deficient essential elements, average daily doses are calculated according to US EPA methodological procedures, but they are modified as necessary because it is a calculation of the health risk from the deficient content of essential elements, not from their increased content, which in turn may be associated with another risk, typically toxic. Health risk calculations use input exposure data in the sense of the US EPA, such as body weight, exposure time and frequency, daily water intake, etc. Input exposure data for the calculation of health risk from deficiency of Ca and Mg in drinking water are summarized in Table 3.

The three defined average daily doses are calculated according to the following equations:ADMD = (MRC − CW) × IR × ED × EF/BW × AT,(2)
ADAD = CW × IR × ED × EF/BW × AT,(3)
ADRD = MRC × IR × ED × EF/BW × AT,(4)
where CW represents the average content of Ca, Mg or average water hardness in the group “soft” water (Table 1), the minimum required concentration (MRC) represents the minimum content of an element, at which there is no known health risk; other abbreviations and parameters used are explained in Table 3. Two methods of the MRC determination are proposed. A very simple way is to determine MRC from drinking water standards, if any exist. The second method is much more complex but at the same time, much more objective. It is based on the actual Ca and Mg contents in drinking water and on the health status of people who drink the respective drinking water. Thus, the actual contents of Ca, Mg and water hardness in the “soft” water group (Table 1) and health indicators, ReC, ReI, ReJ and ReK, of the population supplied with soft drinking water (Table 2) were used to derive MRC.

#### 2.5.1. Derivation of MRC Based on Standard Values 

The Slovak standard for drinking water regulates the values of the three parameters as follows: >30 mg·L^−1^ Ca, 10–30 mg·L^−1^ Mg and water hardness (Ca + Mg) of 1.1–5.0 mmol·L^−1^. Because it is not known how these values were derived, either from health, technical (corrosion, scaling) or a combination of both aspects, it does not seem to be scientifically justifiable to consider these values as MRCs and to calculate ADMD on their basis. Nevertheless, the calculation of ADMD for Mg is provided here, assuming average Mg content of 20 mg·L^−1^:ADRD = 20 × 2 × 1 × 365/70 × 365 = 0.5714 mg·kg^−1^·day^−1^(5)
ADMD = (20 − 5.66) × 2 × 1 × 365/70 × 365 = 0.4097 mg·kg^−1^·day^−1^(6)
ADAD = 5.66 × 2 × 1 × 365/70 × 365 = 0.1617 mg·kg^−1^·day^−1^(7)

From the above calculation, it is clear that ADMD is in principle the difference between ADRD and ADAD. This result (ADMD) would mean that people in soft water areas lack about 0.41 mg·kg^−1^·day^−1^ Mg to achieve ADRD from drinking water. However, as mentioned above, this is just methodical example of calculation and the result is possibly not reliable due to unknown basis of regulatory “MRC”.

#### 2.5.2. Derivation of MRC Based on Real Data

The real Ca and Mg contents, water hardness (Table 1) and values of HIs of the population (Table 2) in the “soft” water group were used to derive MRC. As can be seen in Table 1, “soft” water group is characterized by low Ca and Mg contents in drinking water, which is reflected in worse health status of the population compared with the population supplied with harder drinking water. Although our epidemiological study is only of the ecological type, it was performed on large number of residents and its findings are consistent with meta-analyses of more advanced epidemiological studies on the effect of Ca and Mg in water on cardiovascular mortality. Four main causes of death were taken as the basis for the calculation: ReC, ReI, ReJ and ReK. For comparison, the average national values of the four HIs were used. These national values represent the “no-risk” or “acceptable risk” standard against which the four main causes of death of the population supplied with soft drinking water were compared. The four main causes of death account for about 88% of national mortality: ReC about 25%, ReI ~50%, ReJ ~6% and ReK ~7% [39]. At present (after 2008), the global trend in causes of death slightly increases for OD (to 26%–27%) and moderately decreases for CVD (to 47%–48%). The number of deaths from RS and DS remains virtually unchanged. Other causes of death with the exception of accidental causes of death and suicides, etc., are only insignificant and therefore not considered.

In the first step, optimal Ca and Mg contents and water hardness in drinking water (cross-multiplication) were calculated in order to achieve the average national values of HIs. Each HI was assigned such a weight by which the individual HI contributes to death (see above). In the second step, Ca and Mg contents and water hardness were recalculated according to the weight (according to percentage) of the causes of death and the so-called limit values for Ca, Mg and (Ca + Mg), at which the health status of the population should reach the average national value, were obtained.

According to the declaration of the WHO, the health status of the population is determined by four main factors: (i) lifestyle, (ii) health care, (iii) genetic factors and (iv) environmental factors. To take into account the impact of individual lifestyle or genetic factors (quality and access of health care may be considered as comparable) as well as inter-individual variability on human health, an uncertainty factor (UF) of 2.0 was introduced, and then, the calculated limit values were multiplied with the UF value. This procedure increased the limit values by 100%. In this way, the effect of the other three health determinants is attenuated, and Ca and Mg contents are estimated, at which the four main discussed HIs should be equal to or slightly better than the average national values. The uncertainty factor was introduced mainly for the following reason: The deficient content of Ca and Mg in drinking water probably does not affect all people equally. A higher effect on more sensitive individuals and the lower one on more adaptable individuals can be expected. It is assumed that may affect about half of people, and therefore UF value of 2.0 was chosen. Limit values multiplied with UF represent the minimum required concentration (MRC) (Table 4). The health status of the Slovak population in the four main HIs reaches an average value at this content of Ca, Mg and hardness in drinking water [16,29]. The calculated MRC values of Ca and Mg in drinking water are in accordance with the recommended concentrations of Ca and Mg given by several authors, e.g., [15,17]. They represent the content of elements in drinking water that a person should consume regularly and their health is not endangered. These MRC values were then used as a basis for calculating ADMD, ADRD and ADAD.

Calculation of the three proposed doses, i.e., ADRD, ADMD and ADAD, according to MRC derived from HIs (Table 4), is done as in the previous example (see, Equations (5)–(7)).

#### 2.5.3. Calculation of Hazard Quotient—HQ_d_

The health risk of developing chronic diseases for individual elements/substances is expressed in terms of US EPA [1] as HQ and for concomitant exposure to several elements/substances with the same or similar mechanism of action in the form of Hazard Index (HI). Hazard Index represents the sum of HQ for each element. However, as Ca and Mg are involved in different biochemical reactions and processes in the organism, the calculation of HI does not seem to be appropriate.

According to the US EPA methodology, the following equation is used to calculate HQ:HQ = ADD/RfD,(8)

The equation expresses how much the average daily dose is exceeded and what is the risk expressed in terms of HQ. In our case, the risk is not caused by excess of any harmful element but deficiency of essential elements, so this equation is modified as an inverse ratio of RfD (ADRD) to ADD (ADAD) according to the Equation (1). In this way, it expresses how much lower is the received dose than the required dose and what is the level of health risk in the form of HQ_d_.

An example of the calculation of HQ_d_ is shown for Mg content in drinking water in the “soft” water group. The ADRD and ADAD values derived from the real Mg content in drinking water and HIs of the population in the “soft” water group were used. The reference dose of Mg that a person should take daily from drinking water is the average daily required dose (ADRD), which is 0.476 mg·kg^−1^·day^−1^. The average daily intake (ADAD) is 0.162 mg·kg^−1^·day^−1^. The calculation of HQ_d_ (see Equation (1)) for Mg is as follows:HQ_d_ = 0.476/0.162 = 2.94(9)

More simplified way of calculation, analogous to health risk assessment of airborne contaminants mentioned earlier, is to simply divide MRC (16.66) by CW (5.66), which provides practically the same result of HQ_d_.

The following scale (Table 5) according to US EPA [1] is used to assess the level of risk of chronic disease development.

## 3. Results

The calculated average, median, maximum and minimum values of ADAD and health risk in the form of HQ_d_ based on MRC derived from HIs for both groups of municipalities are given in Table 6. According to the average value of HQ_d_, the population from municipalities with soft drinking water has a medium risk of chronic diseases (HQ_d_ > 1.0, Table 5), while only a low risk of chronic diseases is calculated for the population from municipalities with hard water (HQ_d_ > 0.1, Table 5). The difference in average or median HQ values between the “soft” water group and the “hard” water group is higher than 3-fold. In accordance with the recommendations of US EPA [1], the medium health risk should be notified to the appropriate administrative authorities so that corrective action can be realized, if possible. However, no corrective action is required when the health risk is low or negligible.

More specifically, the municipality with the lowest Mg and Ca contents (Bacúch municipality) has the expected highest values of HQ_dMg_ and HQ_dCa_ (maxima in Table 6 in the “soft” water group), which places it in the category of high risk of chronic diseases (HQ > 4.0, Table 5). Of the 34 municipalities in the “soft” water group, only one municipality (Hronec) is close to low health risk according to HQ_dMg_ (= 0.99, Table 6), while in terms of Ca content, all 34 municipalities are at medium to high health risk because no HQ_dCa_ value was lower than 1.0 (Table 6).

## 4. Discussion

As can be seen from Table 6, the HQ_d_ values for Ca, Mg and (Ca + Mg) are in a narrow range of 2.94–2.95, which corresponds to the medium risk of chronic disease development in the sense of the US EPA. In contrast, meta-analyses of more valid epidemiological studies [15,40,41] result in odds ratio or relative risk of death from cardiovascular disease for water with a higher magnesium content at the level of about 0.75–0.80, which is still a statistically significant risk, but relatively low compared to other known risk factors (for cardiovascular diseases), as it means about a 20%–25% decrease in mortality. However, in the case of high number of cardiovascular diseases in the population, the impact of such relatively low risk factor on public health can be significant. The relatively high level of risk of chronic diseases (HQ_d_ e.g., for Mg = 2.94) fully corresponds to the health status of people supplied with soft drinking water. Mortality from the most common causes of death in Slovakia (cardiovascular and oncological diseases) in the “soft” water group is more than 50% higher compared to the population supplied with hard water. Mortality from the digestive and respiratory systems is two times higher compared to the population supplied with hard water.

The narrow range of three average HQ_d_ (2.94–2.95) is quite surprising. Not for hardness because it is only a function of the sum of Ca + Mg, but surprising is the similarity of HQ_d_ for calcium and magnesium, although both have different functions and effects in the body. The only explanation of this finding is that the effect of magnesium is key for the four diagnoses monitored and its different content in drinking water in the monitored municipalities contributes significantly to the identified health status. However, because the presence of calcium in water is usually associated with magnesium (i.e., soft waters have low contents of both elements, while harder waters are rich in Ca and Mg), the correlation between health status and Ca content in water is similar to that for magnesium. However, any mutual interaction of the two elements cannot be ruled out.

Of all 2037 individual diagnoses listed in the International Classification of Diseases (ICD), 10th revision [33], there was no better HI value in the “soft” water group compared to the “hard” water group. When focusing on individual causes of death, the highest difference in the causes of death was observed for diagnosis *I21*, acute myocardial infarction, 3.23 times higher in the “soft” water group, diagnosis *I25*, chronic ischemic heart disease, 2.17 times higher in the “soft” water group and for diagnosis *C34*, bronchial and lung malignancy, 2.17 times higher in the “soft” water group than in the “hard” water group. As the highest and the most noticeable difference is the number of deaths from the diagnosis of *G80*-infantile cerebral palsy. Only two cases were recorded over the 15 years reviewed in the “hard” water group and up to 34 cases in the “soft” water group. Due to the fact that the cases did not form local or time clusters, it can be assumed that the higher incidence was not caused by any infectious epidemic but another factor or factors, which we hypothesise to be the different hardness of drinking water (see [28]). Clearly, the health status of the population supplied with soft drinking water is much worse than the health status of the population supplied with hard drinking water.

Calcium and magnesium are among the oldest recognized essential elements needed for human health [12]. Adequate amounts of Ca and Mg are required for the proper functioning of human body and maintenance of homeostasis. The recommended daily doses of Ca and Mg for adults are between 700–1200 mg·day^−1^ and 320–420 mg·day^−1^ [42], respectively. Although dairy products are the most prominent source of Ca in the diet, drinking water is another important source of Ca. Water is a major component of the human body and is involved in many body functions, including the transport of nutrients and the removal of waste and toxins. In order to maintain good hydration and body balance, daily intake should reach 1.2 to 2.5 L of water, although needs may vary depending on the age, physical activity or climatic conditions [43]. Since the 1990s, studies have been conducted to assess the bioavailability of Ca contained in calcium-rich mineral water compared to the bioavailability of Ca consumed in dairy products [44,45,46,47,48,49]. These studies show that the bioavailability of Ca from Ca-rich drinking water is equal to or even higher than that from milk and dairy products.

Several epidemiological studies in North America and Europe confirm that children and adults who consume Western type diet have low intake of Mg (30%–50% of the recommended nutritional dose) [50]. For example, 45% of US citizens are at risk of magnesium deficiency and 60% of adults are under ADI (Average Dietary Intake) [51,52,53]. Drinking water is an important source of Mg when it contains Mg at concentrations of around 30 mg·L^−1^ [54].

Calcium and magnesium are found in drinking water of natural origin almost exclusively in the form of real ions and are thus fully accessible to the human body. In various foods, Mg is often bound in the form of complex organic compounds, and therefore its bioavailability to the human body is lower [55]. Under the insufficient intake of these elements by food and their borderline deficit, minor intake from drinking water might play a decisive health role. People who regularly consume drinking water with low Mg content, for example 5 mg·L^−1^, compared to people consuming water with elevated Mg content (e.g., 30 mg·L^−1^) have a daily Mg intake by 50 mg lower, assuming 2 L of water daily. This difference represents approximately 15% of the recommended daily dose. The same is true for Ca.

The effect of Mg deficiency on the increased incidence of cardiovascular diseases was confirmed by several independent meta-analyses [15,40,41]. The meta-analysis of Gianfredi et al. [41] additionally found a statistically significant protective effect of water Ca on cardiovascular diseases. In addition, decreased vascular flexibility and higher arterial age of people drinking soft water were confirmed by Rapant et al. [56]. It was also shown that people drinking Ca and Mg deficient water had a shorter lifespan [16].

The effect of low Mg content and partly also Ca in drinking water on increased mortality from oncological diseases was proved by several studies [55,57,58,59]. The data in our work show that the deficient Ca and Mg contents in drinking water have a significant effect on increased mortality and diseases of the digestive and respiratory system. Although no advanced epidemiological study has yet been performed to directly confirm the relationship to drinking water, the negative effect of insufficient Ca and Mg intake on the state of the immune system, which plays a key role in the etiology of respiratory and digestive diseases, is known. Our ADMD calculation is based on the MRC value derived specifically for the Slovak Republic according to the national epidemiological and health data. This is essentially the preferred approach for health risk assessment by the US EPA method—giving preference to data on the dose-effect relationship and exposure from the same population as the one being assessed. However, this is rather an exception when routinely using the health risk assessment method because such locally produced data are quite rare and mostly dose-response data generated by other studies in other parts of the world have to be taken over. A typical example is the dose-response of arsenic where its limit content in drinking water recommended by the US EPA or WHO was, until recently, based on the half-century-old results of an epidemiological study from Taiwan [60]. Researchers evaluating the Ca and Mg deficiency in water for a population in another part of Europe or the world can therefore use our MRC values but they should be aware of possible differences in drinking water quality, eating habits and health status of the Slovak population.

The Uncertainty Factor value used in this study is also questionable. The value of UF in methodologies for calculating health risks introduced by the US EPA is mostly in the range of 1–10. This factor takes into account all possible uncertainties as well as the possible impact of other health determinants. In our case, UF values of 2 and 3 were taken into account. Based on the results from the application of artificial neural networks [23], UF value of 2 was finally selected. In this way, MRC of Ca and Mg could be determined, at which mortality from the four main causes of death in the Slovak Republic reaches an average. If the UF value of 3 and the subsequently derived MRC were used, the health status would already be better than the average value.

As our assessment revealed a medium, thus relatively serious level of risk, and we are talking about the need for corrective action, the question naturally arises as to what measures are relevant and feasible in this case. The simplest measure seems to recommend to the local population that they should try to compensate the deficit of Ca and Mg from drinking water by changing or modifying the diet. Such a recommendation carries essentially no risk, but it is not possible to estimate how effective it will be. On the one hand, the willingness of the population to change their lifestyle, including nutrition, is not usually high, and on the other hand, we do not even know whether this would actually be reflected in the improvement of health. Our work is based on the assumption that the intake of Ca and Mg by drinking water and its effect are, to some extent, independent of food intake. We also do not know of any intervention study that would confirm the effectiveness of compensating water deficit with food. Therefore, measures should be aimed primarily to increase the content of these elements in water. It can be in the form of recommendation to the population to buy bottled water with higher content of these elements or, still better, to increase the hardness of the supplied drinking water. This can be achieved by selecting a more suitable water source, which may not be available at all in a certain geological area, or by treating the water to increase Ca and Mg contents. Although such treatment is carried out for many soft waters, it is usually done in the simplest and cheapest way, which consists in increasing Ca content in order to reduce the corrosive power of the water. A more suitable technology could also supply Mg, and in addition to the technological function, also fulfil a health function. This is the direction of our next work.

Calcium and magnesium are not regulated as obligatory indicators either under the WHO guidelines or under an EU directive or other international recommendation. In the currently completed revision of the EU Directive 98/83/EC, at least one qualitative requirement was added to replenish the content of minerals in water, the content of which was significantly reduced due to the treatment or conditioning: “This applies particularly to waters undergoing treatment (demineralization, softening, membrane treatment, reverse osmosis, etc.). Where water intended for human consumption is derived from treatment that significantly demineralizes or softens water, calcium and magnesium salts could be added to condition the water in order to reduce possible negative health impact, as well as corrosion or aggression of water and to improve taste. Minimum concentrations of calcium and magnesium or total dissolved solids in softened or demineralized water could be established taking into account the characteristics of water that enters these processes.” [61].

Of the 28 EU countries, the minimum content of Ca, Mg or hardness in national standards as a minimum recommended value is regulated in only 11 countries, with varying degrees of binding force [62]. In terms of the results of the present study, probably the most correct standard for drinking water is in the Czech Republic. The Czech standard recommends minimum Mg and Ca contents of 10 mg·L^−1^ and Ca 30 mg·L^−1^, respectively. However, this applies only to water in which Mg and Ca contents are reduced by the treatment. In addition, the standard recommends ranges of the following elements as an optimum in terms of health impact: Mg 20 up to 30 mg·L^−1^ and Ca 40 up to 80 mg·L^−1^ [62]. At least, these Ca and Mg contents in drinking water can be recommended to keep the population at the lowest possible health risk.

## 5. Conclusions

Although most of the required daily intake of essential elements is covered by food, drinking water can make a significant contribution in some cases, which in the case of lower food intake plays a decisive role in the development of clinical signs of deficiency. In addition, it seems that the intake of essential elements with drinking water also has a special qualitative significance and their absence or low contents can have a negative effect on the human health even when they are present in the diet at sufficient levels. People regularly consume the same drinking water, especially in their place of residence. Therefore, in the case of low content of essential elements (e.g., Ca and Mg) in drinking water, their deficit gradually increases because Ca and Mg contents in cooked food are also lower due to higher leaching and losses [63], which may be reflected in their deteriorated health.

Therefore, we propose to supplement the US EPA’s health risk assessment for harmful elements or substances with a risk analysis for deficient essential elements. Proposed methodology allows one to evaluate and quantify possible adverse human health effects due to deficient content and intake of essential elements. The proposed methodology, once verified by independent research teams, can be a very important tool for the protection of human health and this represents a basic precondition for identifying the need for corrective action and consequently reducing the negative impacts on the human health. Based on the results of our risk analysis, it was found that the population supplied with soft drinking water is at medium risk. The results of risk analysis were confronted directly with the health status of the population supplied with soft drinking water. Data on the health status of the population are relatively extensive. They represent approximately 1.6 million person years, and therefore, the results obtained can be considered as supported by sufficiently large data set, even when the epidemiological study carried out is only of an ecological type.

Based on the results of this work, it is possible to recommend WHO to include the content of Ca and Mg among the regulated parameters in the standard for drinking water and thus adjust not only their upper limit, but also the minimum content. The use of the proposed method of risk analysis for deficient essential elements can be applied to other essential elements and to other components of the environment.

## Figures and Tables

**Figure 1 ijerph-17-05915-f001:**
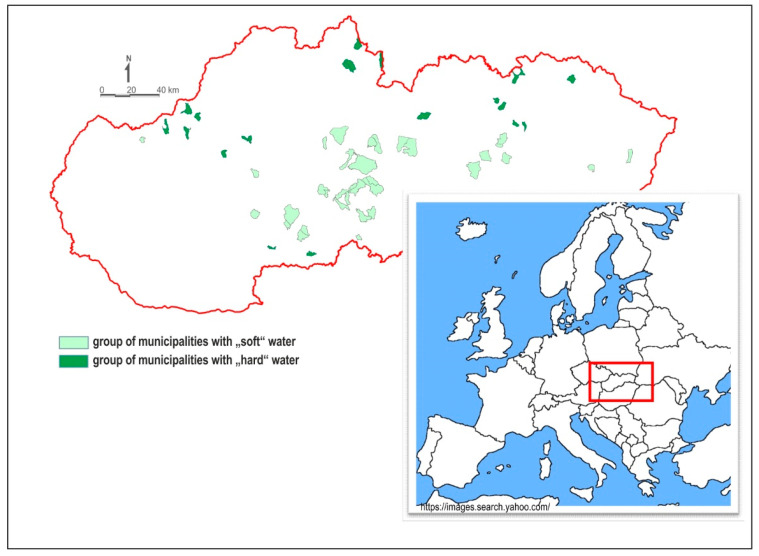
Schematic map showing location of the Slovak Republic in Europe and municipalities divided into the two groups: group of municipalities with “soft” water and group of municipalities with “hard” water.

**Table 1 ijerph-17-05915-t001:** Statistical summary of Ca and Mg contents and water hardness in “soft” and “hard” water groups.

Group of Municipalities		Ca	Mg	(Ca + Mg)
		mg·L^−1^	mg·L^−1^	mmol·L^−1^
“soft” water	Average	21.2	5.66	0.77
	Minimum	5.40	2.00	0.26
	Maximum	48.9	16.8	1.69
	Median	20.7	4.45	0.75
“hard” water	Average	70.1	26.4	2.84
	Minimum	38.5	12.6	1.58
	Maximum	96.8	38.0	3.84
	Median	73.7	27.6	2.75

**Table 2 ijerph-17-05915-t002:** Values of the main health indicators. LE is given in years and ReC, ReI, ReJ, ReK as the number of cases per 100,000—standard SR %.

Group of Municipalities	LE ^1^	ReC ^2^	ReI ^3^	ReJ ^4^	ReK ^5^
“soft” water	69.40	282.53	852.42	100.92	82.65
“hard” water	74.68	171.11	455.11	45.37	34.27
Slovak Republic	72.60	212.79	531.05	58.08	45.83

Note: data are recalculated to the number of residents in individual municipalities and represent the Bayesian average. ^1^ life expectancy at birth calculated for a period of 1994–2008; ^2^ relative mortality from malignant tumor (number of deaths per 100,000 residents) calculated for a period of 1994–2008; ^3^ relative mortality from diseases of the circulatory system (number of deaths per 100,000 residents) calculated for a period of 1994–2008; ^4^ relative mortality from respiratory diseases (number of deaths per 100,000 residents) calculated for a period of 1994–2008; ^5^ relative mortality from diseases of the digestive system (number of deaths per 100,000 residents) calculated for a period of 1994–2008.

**Table 3 ijerph-17-05915-t003:** Input exposure data for the estimation of health risk from deficient Ca and Mg contents in drinking water for adult population.

Parameter	Value	Unit	Source
BW—body weight	70	kg	[2]
AT—averaging exposure time	365	day	[38]
CW—content of chemical element in water	site specific	mg·L^−1^	
IR—daily water intake	2	L·day^−1^	[38]
EF—exposure frequency	365	day·year^−1^	[1]
ED—duration of exposure	1	year	[38]

**Table 4 ijerph-17-05915-t004:** Limit and minimum required Ca and Mg contents and water hardness (Ca + Mg) in drinking water derived from health indicators (ReC, ReI, ReJ and ReK) of the “soft” water group and national average for “soft” water group.

Calcium						
Mean “soft” water	Ca_ReC_ ^1^	Ca_ReI_	Ca_ReJ_	Ca_ReK_	Ca	
					LV ^2^	MRC ^3^
	28.15	34.03	36.84	38.23	31.21	62.42
**Magnesium**						
Mean “soft” water	Mg_ReC_	Mg_ReI_	Mg_ReJ_	Mg_ReK_	Mg	
					LV	MRC
	7.52	9.09	9.83	10.21	8.33	16.66
**Water Hardness**						
Mean “soft” water	(Ca + Mg)_ReC_	(Ca + Mg)_ReI_	(Ca + Mg)_ReJ_	(Ca + Mg)_ReK_	(Ca + Mg)	
					LV	MRC
	1.02	1.24	1.34	1.39	1.13	2.26

^1^ limit values (LV) of Ca and Mg contents (mg·L^−1^) and water hardness (mmol·L^−1^) for individual health indicators calculated on the basis of average Slovak nationwide values of health indicators; ^2^ limit values for Ca, Mg and (Ca + Mg) contents calculated as a weighted average of the limit values of Ca, Mg and (Ca + Mg) for individual health indicators according to their weight; ^3^ minimum required concentration of Ca, Mg and (Ca + Mg) calculated as the weighted average of the limit content of Ca, Mg and (Ca + Mg) for individual health indicators according to their weight increased by the uncertainty factor (a value of 2.0).

**Table 5 ijerph-17-05915-t005:** Scale for classifying the level of risk of chronic diseases.

Risk Level	HQ_d_	Risk of Chronic Diseases
1	≤0.1	Without risk
2	>0.1	Low
3	>1.0	Medium
4	>4.0	High

**Table 6 ijerph-17-05915-t006:** Calculated HQ_d_ values according to MRC derived from health indicators.

	ADAD_Mg_ ^1^	ADAD_Ca_	ADAD_(Ca + Mg)_	HQ_dMg_ ^2^	HQ_dCa_	HQ_d(Ca + Mg)_
“soft” water group						
Average	0.162	0.606	0.022	2.94	2.94	2.95
Median	0.127	0.591	0.021	3.75	3.01	3.01
Maximum	0.480	1.397	0.048	8.33	11.6	8.69
Minimum	0.057	0.154	0.007	0.99	1.28	1.34
“hard” water group						
Average	0.754	2.003	0.081	0.63	0.89	0.80
Median	0.789	2.106	0.079	0.60	0.85	0.82
Maximum	1.086	2.766	0.110	1.32	1.62	1.43
Minimum	0.360	1.100	0.045	0.44	0.64	0.59

^1^ ADAD in mg·kg^−1^·day^−1^; ^2^ hazard quotient.

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
