# Peer review of "Proposal of New Health Risk Assessment Method for Deficient Essential Elements in Drinking Water—Case Study of the Slovak Republic"

_ijerph, 2020, doi:10.3390/ijerph17165915_

Round 1

Reviewer 1 Report

The results in 4 lines is probably not enough. You should discuss and compare the results obtained with soft and hard water

The summary should be strictly in relation to the title.

other comments in the text

Author Response

Dear Reviewer 1,

Thank you very much for your valuable comments on our manuscript. We think that all your comments were helpful in improving the manuscript.

Our responses to your comments are listed below and changes made in the manuscript are shown by “Track changes” function.

We hope that the changes in the manuscript will be sufficient for you.

Comment 1, (Lines 321–323) - No sample calculations are needed, they have already been repeated (Line 263)

Response: You are fully right that sample calculation repeat here, and therefore, they seem to be superfluous. Therefore, they were deleted and replaced by the sentence:

Calculation of the three proposed doses, i.e. ADRD, ADMD and ADAD, according to MRC derived from HIs (Table 4), is done as in the previous example (see, equations 5, 6 and 7).

Comment 2, Results - The results in 4 lines is probably not enough. You should discuss and compare the results obtained with soft and hard water.

Response: Thank you very much for your comment. We extended the results and compared the results between the two groups of municipalities. The results in the revised manuscript appear as follows:

The calculated average, median, maximum and minimum values of ADAD and health risk in the form of HQd based on MRC derived from HIs for both groups of municipalities are given in Table 6. According to the average value of HQd, the population from municipalities with soft water has a medium risk of chronic diseases (HQd >1.0, Table 5), while only a low risk of chronic diseases is calculated for the population from municipalities with hard water (HQd >0.1, Table 5). The difference in average or median HQ values between the “soft” water group and the “hard” water group is more than 3-fold. In accordance with the recommendations of US EPA [1], the medium health risk should be notified to the appropriate administrative authorities so that corrective action can be realized, if possible. However, no corrective action is required when the health risk is low or negligible.

More specifically, the municipality with the lowest Mg and Ca contents (Bacúch municipality) has the expected highest values of HQdMg and HQdCa (maxima in Table 6 in the “soft” water group), which places it in the category of high risk of chronic diseases (HQ >4.0, Table 5). Of the 34 municipalities in the “soft” water group, only one municipality (Hronec) is close to low health risk according to HQdMg (= 0.99, Table 6), while in terms of Ca content, all 34 municipalities are at medium to high health because no HQdCa value was lower than 1.0 (Table 6).

Comment 3, tab. 6 - Table 6 should be in the results, increased by all recalculations and not based only on average results and only on soft water. There is a lack of many results in this table, from all the positions in Figure 1. It would be worth pointing out the places with the highest and lowest HQs and try to find out what influenced the discussion.

Response: Thank you again for your valuable comments. We extended Table 6. Revised form of this table was moved into “Results” section and enriched in calculations of average, median, minimum and maximum values of ADAD and HQd for Ca, Mg and (Ca + Mg), i.e. water hardness. The results shown in Table 6 were then commented in “Results” section of the revised manuscript.

Comment 4, (Lines 424–425) - Ca and Mg affect water hardness. In contrast, hard water is healthier to drink when there are harmful substances, e.g. heavy metals. Their harmfulness depends to a large extent on the hardness of the water. So not so directly, only Ca and Mg determine the life expectancy.

Response: Dear reviewer, you are right. However, in our case, it is drinking water, whose microbiological and chemical quality in terms of the content of harmful elements and substances is strictly controlled. These drinking waters do not have increased levels of harmful substances, including heavy metals. Therefore, possible effect of the presence of harmful elements and substances on the life expectancy would be largely negligible.

Comment 5, (Lines 476–482) - I don't know if you need to quote this passage

Response: We think that the cited statement of the EU Directive on the health effects of soft and hard water legislatively supports the achieved results of the article and we therefore propose to keep it.

Comment 6, (Lines 493–501) - I think it's superfluous. The summary should be strictly in relation to the title.

Response: We fully agree with the reviewer, and therefore, we removed the first three sentences of the conclusion accordingly.

Sincerely Yours,

Stanislav Rapant and co-authors.

Reviewer 2 Report

The authors addressed the reviewer's comments adequatly. I suggest the article be published. 

Author Response

Dear Reviewer 2,

Many thanks for your positive response to our revised version of the manuscript.

You have only concern, regarding English language. We sent the manuscript to a person from UK through our family members, living in UK, and she found no significant errors in the manuscript. She had to correct only some very minor errors.

Again, thank you very much for your positive support and time spent at reviewing our study.

Sincerely Yours,

Stanislav Rapant and co-authors.

Reviewer 3 Report

Some of my original comments were not address. I do not have additional comments. 

Author Response

Dear reviewer 3,

Thank you very much for your time spent reviewing the manuscript. We really appreciate your positive support of our study.

Sincerely Yours,

Stanislav Rapant and co-authors.
